# Influences of Digital Literacy and Moral Sensitivity on Artificial Intelligence Ethics Awareness Among Nursing Students

**DOI:** 10.3390/healthcare12212172

**Published:** 2024-10-31

**Authors:** Yaki Yang

**Affiliations:** Department of Nursing, Wonkwang University, Iksan-si 54538, Republic of Korea; ykyang@wku.ac.kr

**Keywords:** literacy, artificial intelligence, awareness, nursing, students

## Abstract

Background: As artificial intelligence technology has developed, research on the application of AI in the medical field has increased, and there is a high likelihood that the use of AI technology will expand in nursing education and practice in the future. However, ethical issues arise when utilizing AI, necessitating a high level of ethical awareness before its application. Objectives: This study aimed to identify factors in artificial intelligence ethics awareness among nursing students. Methods: Participants were 140 nursing students attending universities in G City and J Province in South Korea. Data were collected using a self-administered questionnaire from 26 August to 6 September 2024. Factors in artificial intelligence ethics awareness were analyzed by multiple regression analysis. Results: Nursing students’ artificial intelligence ethics awareness is significantly correlated with digital literacy (r = 0.30, *p* < 0.001) and moral sensitivity (r = 27, *p* < 0.001). The influencing factor in artificial intelligence ethics awareness among nursing students was moral sensitivity (β = 0.23, *p* = 0.042). The explanation power of these variables was 14.0% (F = 46.78, *p* < 0.001). Conclusions: There is a need to provide education and training programs that can improve moral sensitivity to foster artificial intelligence ethics awareness.

## 1. Introduction

Recently, with the widespread use of AI (Artificial Intelligence)-based devices in various aspects of daily life, applications in the healthcare sector have emerged, including clinical diagnosis, disease prediction, imaging interpretation, surgical robots, and app-based medical services. While there is a positive perspective that AI will make human tasks more convenient in various fields, there are also concerns related to ethical issues. In particular, in the healthcare sector, ethical issues may arise regarding data bias, safety, privacy violations, and legal liability for damages resulting from inaccurate diagnoses or prescriptions [1].

A study using an AI app to predict the likelihood of patients developing complications after being diagnosed with pneumonia and hospitalizing them resulted in the incorrect decision to send asthma patients home rather than hospitalizing them [2]. This illustrates that it is difficult to fully trust the judgments made by AI-based devices, as they may produce inaccurate diagnoses. Ethical conflicts are inevitable in the use of AI, so it is essential to cultivate a high level of ethical awareness to reduce the likelihood of ethical issues arising.

In South Korea, the Ministry of Science and ICT (Information & Communication Technology) announced a human-centered national AI ethics guideline in 2020, emphasizing the importance of ethical awareness in AI. As a result, research related to AI ethics education has been actively pursued [3]. If awareness of AI that will coexist with our society is low and there is no appropriate ethical response plan, it can cause confusion not only for medical professionals but also for patients and their caregivers, so it is urgent to come up with countermeasures. Therefore, fostering ethical awareness related to AI is of utmost importance, and there is a need for contemplation on bioethics and AI ethics, as well as ethical decision-making skills to address the various issues that may arise with the introduction of AI technology [4].

Currently, the application of AI technology in nursing tasks is not widespread. General university students, including nursing students, have had experiences using AI-based devices in real life [5], and active research is being conducted on the educational use of AI technology in the academic field [6,7,8]. In particular, the education sector effectively utilizes AI technology for language education by employing the natural language processing capabilities of generative AI chat bots, allowing users to learn natural conversation patterns [9]. Moreover, universities are recognizing the importance of AI-related education, leading to an expansion of AI-related courses [10,11]. Some domestic universities are offering general education courses related to understanding and utilizing AI as part of their AI and software-centered university programs to foster an academic perspective on AI [3].

Given the potential for AI technology to be applied in nursing education and practice in the future, it is crucial to provide AI ethics education for nursing students to prepare for the confusion that may arise from its application. Additionally, nursing educators should guide students in forming correct values regarding the application of AI. Before implementing such education, it is essential to explore the factors that influence nursing students’ awareness of AI ethics.

Digital literacy refers to the individual capacity to search, organize, and utilize information in a digital knowledge-based society [12]. The Digital Literacy Education Association defines it as a competency that goes beyond simple information usage skills, aiming to foster a healthy social culture and promote balanced development for individuals and society through the use of digital technologies. This definition extends to encompass the concept of digital citizenship [13]. Additionally, digital literacy is a comprehensive concept that includes ICT literacy, technology literacy, and AI literacy [14]. Prior research indicates that computational thinking, which involves using computers to solve problems, should be the foundation for AI utilization education [15]. Therefore, it is crucial to assess the level of digital literacy in the age of AI.

While there are studies that have examined the digital literacy of nursing students, research exploring the impact of their digital literacy on awareness of AI and ethical consciousness is currently lacking.

Moral sensitivity is the ability to recognize the existence of moral issues in real-life situations and is a core component of moral judgment capacity [16]. Although there has been no prior research specifically addressing the relationship between moral sensitivity and AI ethics awareness, studies have emphasized the necessity of AI ethics education to enhance moral sensitivity and judgment as essential ethical competencies for the AI era [17]. Additionally, research has shown that moral sensitivity, along with critical thinking disposition, influences nurses’ awareness of ethical codes [18]. Based on these findings, it is essential to conduct studies focusing on the relationship between moral sensitivity and AI ethics awareness among nursing students.

Therefore, this study aims to examine the relationships between digital literacy, moral sensitivity, and AI ethics awareness among nursing students in the era of the Fourth Industrial Revolution, where AI ethics is emphasized. Additionally, it seeks to explore how digital literacy and moral sensitivity affect AI ethics awareness. The findings will serve as foundational data for the development and provision of educational programs aimed at fostering AI ethics awareness.

The purpose of this study is to investigate the effects of digital literacy and moral sensitivity on AI ethics awareness among nursing students. The specific objectives are to assess digital literacy, moral sensitivity, and AI ethics awareness and to examine the relationships between digital literacy, moral sensitivity, and AI ethics awareness among the participants. Additionally, the study aims to identify the factors influencing AI ethics awareness among the participants.

## 2. Materials and Methods

### 2.1. Study Design and Sample

This study is a descriptive research investigation conducted to assess the digital literacy, moral sensitivity, and AI ethics awareness of nursing students, as well as to identify the factors influencing their AI ethics awareness. The subjects of this study are 140 nursing students enrolled at comprehensive universities in G City and J Province in South Korea, who understood the purpose of the research and voluntarily agreed to participate. The required number of participants was calculated using the G*Power 3.1.9.4 program. The required sample size for the independent samples *t*-test (effect size = 0.80, significance level = 0.05, power = 0.80, two-tailed test) was 52. The required sample size for correlation analysis (effect size = 0.30, significance level = 0.05, power = 0.80, two-tailed test) was 82. The required sample size for multiple regression (effect size = 0.15, significance level = 0.05, power = 0.80, two-tailed test) was 109. Considering the dropout rate, a total of 150 individuals were selected as research subjects. After excluding those who provided inconsistent responses, a total of 140 participants were included in the final analysis.

### 2.2. Instrument

A structured self-administered questionnaire was utilized, comprising a total of 84 items categorized into 6 items on general characteristics, 18 items on digital literacy, 36 items on moral sensitivity, and 24 items on AI ethics awareness. Before using the research instruments, approval for their use was obtained.

#### 2.2.1. Digital Literacy

To measure digital literacy, this study utilized the digital literacy assessment tool [19]. This tool consists of a total of 18 items and employs a 5-point Likert scale, with a possible score range from 18 to 90. Higher scores indicate a higher level of digital literacy. The items are categorized as follows: 5 items related to basic ICT competencies, 5 items related to social media use and collaboration skills, 4 items assessing basic task utilization skills, and 4 items measuring adaptability to a software-centric society. The reliability of the tool at the time of development was Cronbach’s α = 0.91, while in this study, it was Cronbach’s α = 0.93.

#### 2.2.2. Moral Sensitivity

Moral sensitivity was measured using the Ethical Sensitivity Scale for Clinical Nurses [20]. This tool consists of a total of 36 items and utilizes a 5-point Likert scale, with a possible score range from 36 to 180. Higher scores indicate greater moral sensitivity. The items are categorized as follows: 6 items related to patient respect, 6 items related to professional ethics, 6 items assessing nursing responsibilities, 5 items concerning the intention to act, 4 items focused on ethical reflection, 3 items regarding ethical burden, 3 items related to ethical situation awareness, and 3 items addressing empathy. The reliability of the tool at the time of development was Cronbach’s α = 0.92, while in this study, it was Cronbach’s α = 0.88.

#### 2.2.3. AI Ethics Awareness

AI ethics awareness was measured using the Test for Artificial Intelligence Ethics Awareness developed by Kim et al. for elementary, middle, and high school students [21]. This tool consists of a total of 24 items and employs a 5-point Likert scale, with a possible score range from 24 to 120. Higher scores indicate greater awareness of AI ethics. The items are categorized as follows: 3 items related to responsibility, 3 items related to stability, 3 items addressing non-discrimination, 3 items regarding transparency and explainability, 3 items focused on human-centered services, 3 items concerning employment, 3 items about permissions and limitations, and 3 items related to robot rights. The reliability of the tool at the time of development was Cronbach’s α = 0.81, while in this study, it was Cronbach’s α = 0.74.

### 2.3. Data Collection

Data collection for this study was conducted at comprehensive universities located in G city and J Province from 26 August to 6 September 2024. Prior to data collection, permission was obtained from the nursing department of each university after explaining the purpose, participants, and methods of the study. Participants were recruited voluntarily through recruitment notices. For those capable of self-reporting, they were asked to complete the questionnaire independently. The average time required to complete the questionnaire was approximately 10 to 15 min.

### 2.4. Ethical Consideration

This study was conducted after obtaining approval from the Institutional Review Board (IRB) (Approval Number: 2024-08-SB-053) prior to data collection. Participants were informed about the purpose, methods, expected outcomes, and assurances of anonymity of the study to protect their personal information. It was clarified that their data would not be used for any purposes other than research. Participants were also provided with an explanation sheet stating that there would be no disadvantages for non-participation or withdrawal from the study. The questionnaires were collected by the researchers directly and were anonymously coded. The completed questionnaires were stored in a secure cabinet. After the conclusion of the study, the questionnaires will be kept for 3 years and then securely disposed of.

### 2.5. Statistical Analysis

Data analysis was conducted using SPSS/WIN 28.0 software. Descriptive statistics were used to analyze the general characteristics of the participants. Digital literacy, moral sensitivity, and AI ethics awareness among the participants were calculated using means and standard deviations. Average scores of each variable show the value obtained by dividing the total score by the number of items. Differences in AI ethics awareness based on general characteristics were assessed using the independent samples *t*-tests and Analysis of Variance (ANOVA). If a significant difference was found in ANOVA, a further study of Scheffé was conducted. The correlation between measured variables was examined using Pearson’s correlation coefficient. Factors influencing AI ethics awareness among the participants were analyzed using multiple regression analysis.

## 3. Results

### 3.1. Participant Characteristics and Differences in AI Ethics Awareness

This study included a total of 140 participants, with females constituting 93.6% of the sample. The largest group was second-year students, accounting for 47.1%. Regarding religion, 60.7% reported having a religion. In terms of experience using artificial intelligence devices, 75.0% indicated they had such experience, while 92.9% had undergone ethics-related education and 67.9% had experience with AI-related education. AI ethics awareness among the participants showed significant differences based on religion (t = −1.98; *p* = 0.025) and experience with AI-related education (t = 2.85; *p* = 0.003) (Table 1).

### 3.2. Degrees of Digital Literacy, Moral Sensitivity, and AI Ethics Awareness

Digital literacy had an average score of 4.09. Moral sensitivity had an average score of 4.25. When examining subdomains, the area of “respect for patient” scored the highest, while the area of “willingness to do good” scored the lowest. AI ethics awareness had an average score of 3.39. In terms of specific subdomains, the area of “transparency and explainability” had the highest score (Table 2).

### 3.3. Correlation Among Digital Literacy, Moral Sensitivity, and AI Ethics Awareness

The analysis of the correlations among digital literacy, moral sensitivity, and AI ethics awareness revealed that nursing students’ AI ethics awareness is significantly correlated with digital literacy (r = 0.30, *p* < 0.001) and moral sensitivity (r = 0.27, *p* < 0.001). This indicates that higher levels of digital literacy and moral sensitivity are associated with greater AI ethics awareness (Table 3).

### 3.4. Factors Influencing AI Ethics Awareness

To identify the factors influencing AI ethics awareness among the participants, a multiple regression analysis was conducted, as shown in Table 4. The independent variables included religion, experience in AI-related education that had statistically significant differences in AI ethics awareness based on general characteristics, digital literacy, and moral sensitivity. Religion and experience in AI-related education were dummy coded, with “no religion” and “having AI-related education experience” designated as the reference groups. The Durbin–Watson statistic was 1.545, indicating that the assumption of residual independence was met. The Variance Inflation Factor (VIF) values were below 10, suggesting that there were no issues with multicollinearity. The regression model in this study was statistically significant (F = 46.78, *p* < 0.001). The primary factor influencing AI ethics awareness among nursing students is moral sensitivity, with the explanatory power of the variables being 14.0%. The variable that had a statistically significant effect on AI ethics awareness is moral sensitivity (β = 0.23, *p* = 0.042).

## 4. Discussion

This study was conducted to analyze the relationships between digital literacy, moral sensitivity, and AI ethics awareness among nursing students, as well as the influence of digital literacy and moral sensitivity on AI ethics awareness. The aim was to provide foundational data to foster a positive environment for enhancing AI ethics awareness among nursing students. Based on the research findings, the following points will be discussed:

The digital literacy of nursing students was found to be at an average level of 4.09 out of 5, indicating a score above the midpoint. This level is similar to the findings, which used the same tool for nursing students [22], as well as to other studies that assessed digital literacy among nursing students using different instruments [23]. The reason why the level of digital literacy in this study and previous studies was high is because the research subjects were mostly young nursing students in their 20s, so it is believed that the level of digital literacy is high. Supporting this, a study found that young adults have relatively higher levels of digital media literacy compared to children and older age groups [24].

Moral sensitivity among nursing students was assessed at an average score of 4.25 out of 5, indicating a level above the midpoint. This score is higher than those reported in studies [25,26], which used the same measurement tool for clinical nurses. While clinical nurses are required to demonstrate high levels of moral sensitivity in their practice, they often face ethical dilemmas that can expose them to conflicts, resulting in lower levels of moral sensitivity compared to nursing students [27].

Research indicates that nursing students have higher levels of bioethical awareness compared to non-nursing students [28], and that there is a positive correlation between bioethical awareness and moral sensitivity [29]. Thus, while nursing students may exhibit relatively high levels of moral sensitivity, the challenges faced in clinical practice, especially in ethically complex situations, may lead to a decline in their moral sensitivity once they become clinical nurses.

Given that 98.6% of clinical nurses reported difficulty in seeking concrete advice for problem solving in ethical dilemmas and expressed the need for clinical ethics consultation services [30], it is essential to understand the challenges nurses encounter in these situations. There is a pressing need for policy support to provide ethical resources that can help address these challenges.

AI ethics awareness among nursing students was found to be 3.39 out of 5. When examining the subdomains, the scores were highest for transparency and explainability, followed by employment, responsibility, person-centered services, robot rights, permissible limits, stability, and non-discrimination. This level of awareness is consistent with results from a study [31] that used the same measurement tool for nursing students, and it is higher than that reported for clinical nurses in a study [32].

In particular, the “transparency and explainability” subfactor scored the highest in this study, aligning with the findings [31]. Overall, the AI ethics awareness among nursing students was above average, indicating that they place significant importance on the explainability of decision-making processes and the disclosure of potential risks when utilizing AI technologies.

Future research should focus on examining whether AI-based devices in clinical settings meet the standards of transparency and explainability.

The differences in AI ethics awareness based on general characteristics showed significant variations according to religion and experience with AI-related education. Contrary to the findings [33], which indicated that individuals with religious beliefs exhibited significantly higher levels of bioethical awareness, this study found different results. While there are claims that non-religious individuals may lack ethical awareness [34], some studies suggest that personal religious practices do not necessarily influence ethical awareness [35]. This highlights the complexity of the relationship between religious affiliation and ethical awareness, indicating a need for repeated studies with different populations and regions.

Furthermore, this study found that individuals with experience in AI-related education had significantly higher levels of AI ethics awareness. This is supported by research indicating that educational programs emphasizing AI ethics positively impact the enhancement of ethical awareness [36]. Therefore, it is essential to develop and implement educational programs related to AI ethics to further improve awareness in this area.

The results of this study indicated a statistically significant positive correlation between AI ethics awareness and digital literacy, confirming that higher levels of digital literacy are associated with greater AI ethics awareness. There is a lack of research specifically examining the correlation between these variables; however, similar studies have shown a significant positive correlation between information competence and information ethics awareness [37], supporting this finding. Additionally, research investigating the relationship between college students’ digital media literacy and moral sensitivity, as well as bioethical awareness, also found a positive correlation between digital media literacy and bioethical awareness [38], which aligns with our results. This suggests that individuals who are proficient in using digital devices and have a higher level of knowledge in this area tend to exhibit greater AI ethics awareness. Therefore, enhancing knowledge of digital device usage is crucial for fostering AI ethics awareness in the information age.

Moreover, AI ethics awareness was also found to have a significant positive correlation with moral sensitivity, indicating that higher levels of moral sensitivity correspond with greater AI ethics awareness. Although there is limited research directly addressing the correlation between these specific variables, related studies have found a positive relationship between moral sensitivity and ethical competence among nurses [39]. Additionally, a positive correlation was observed between moral sensitivity and bioethical awareness among nursing students [29], which aligns with our findings. This indicates that as individuals demonstrate more ethical behavior, their awareness of AI-related ethics also increases. Therefore, to enhance AI ethics awareness, it is essential to strengthen the ethical competence of the target population.

The regression analysis indicated that the primary factor influencing AI ethics awareness among nursing students is moral sensitivity, with the explanatory power of the variables being 14.0%. There is a limited amount of research directly addressing the relationship between moral sensitivity and AI ethics awareness; however, similar studies have shown that ethical identity significantly impacts internet ethics awareness, aligning with our findings [40]. Additionally, a study found that privacy sensitivity and ethical identity significantly influenced internet ethics awareness, further supporting our results [41].

The relatively low explanatory power of 14.0% in this study may be attributed to the negative attitudes toward AI observed among the participants. Previous research has indicated that moral sensitivity has a significant positive effect on bioethical awareness [29]. Furthermore, students and professionals in healthcare fields often display a negative tendency toward AI due to concerns about job displacement, suggesting that higher levels of bioethical awareness may correlate with negative attitudes towards AI [4]. Similarly, a study found that higher levels of bioethical awareness were associated with negative attitudes toward AI, leading to lower AI ethics awareness [32]. However, since this study did not assess the participants’ perceptions and attitudes toward AI or their levels of bioethical awareness, it is difficult to generalize these conclusions.

Bioethical awareness is a critical examination of the moral dimension of the decision-making process regarding human life and death [42]. In the healthcare field, various ethical issues including moral dimensions may arise when artificial intelligence is used in the treatment process. Therefore, there is a need for future research to identify the relationship between awareness and attitude toward artificial intelligence, bioethical awareness, and artificial intelligence ethical awareness. While this study identified moral sensitivity as a significant factor influencing AI ethics awareness, the low level of explanatory power suggests that additional factors affecting nursing students’ AI ethics awareness need to be identified. Previous studies have identified various factors influencing bioethical awareness among nursing students, including ethical values, experiences with bioethical issues, and the quality and quantity of bioethics education within their curricula [43,44]. These factors showed higher explanatory power compared to this study. Therefore, future research should investigate the ethical characteristics of participants and their relationship with AI ethics awareness, as well as other influences on AI ethics awareness.

In this study, while ethics education did not significantly impact AI ethics awareness, previous research emphasizes the necessity of AI ethics education to cultivate autonomous agents with personal characteristics in a generation where AI technology is pervasive [45]. Furthermore, studies assessing AI ethics awareness among university students have highlighted negative aspects of technological advancement, such as privacy invasion, misuse of technology, and information bias, underscoring the need for education related to AI ethics to mitigate these issues [15]. When synthesizing these findings, it is evident that there is a high demand for AI ethics education. Given that nursing students largely acquire bioethical knowledge through their coursework [46], it is essential to provide ethics education starting from the first or second year to help them establish a solid ethical framework before beginning clinical practice. Additionally, research to verify the effectiveness of such educational programs is necessary to ensure that the curriculum meets the ethical demands of future healthcare professionals. This approach will not only enhance AI ethics awareness among nursing students but also prepare them to navigate ethical challenges in a technology-driven healthcare environment.

The results of this study confirmed that moral sensitivity significantly influences AI ethics awareness. Previous research examining the effectiveness of nursing ethics education has shown that such education is categorized into bioethics and professional ethics training. Students who received nursing ethics education demonstrated significantly higher levels of moral sensitivity and bioethics awareness [47]. Additionally, a study focusing on the effects of bioethics education on nursing students found that those who participated in bioethics training exhibited a significant increase in their levels of moral sensitivity [48].

Based on the findings that moral sensitivity significantly impacts AI ethics awareness, it is essential to implement nursing ethics education aimed at enhancing moral sensitivity, with a particular emphasis on integrating bioethics-related content. Furthermore, research should be conducted to determine whether bioethics education effectively improves AI ethics awareness. This focus will ensure that nursing students are better equipped to navigate the ethical challenges presented by AI technologies in clinical settings, ultimately contributing to more ethically responsible healthcare practices.

## 5. Conclusions

The results showed that the participants’ AI ethics awareness significantly differed based on general characteristics such as religion and experience with AI-related education. Additionally, AI ethics awareness was found to have a significant positive correlation with digital literacy and moral sensitivity. The influencing factor in participants’ AI ethics awareness was identified as moral sensitivity, with the explanatory power of the variables being 14.0%. These findings can serve as foundational data for the development and implementation of educational programs aimed at enhancing AI ethics awareness among nursing students in the future. Based on the above results, the following recommendations are proposed. First, while this study confirmed that moral sensitivity affects nursing students’ AI ethics awareness, the explanatory power was only 14.0%, indicating a low level. Considering previous studies that assessed the level of ethical awareness based on general and ethical characteristics, it is suggested that follow-up research examine the impact of ethical characteristics, as well as the timing and frequency of ethics-related education experienced by participants, on AI ethics awareness. Second, since moral sensitivity can be enhanced through nursing ethics education and bioethics education has been shown to be effective in improving moral sensitivity, it is recommended that any ethical education programs developed for nursing students include content related to bioethics. Therefore, it is necessary to check whether the test–retest reliability is stable; even if the size of the subject and the hospital sizes in the final selection of items reliably measure test–retest reliability, it is necessary to check these.

## 6. Strength and Limitations

In the era of the Fourth Industrial Revolution, where AI ethics awareness is emphasized, the significance of this research lies in identifying the factors influencing the AI ethics awareness among nursing students, who will be future healthcare professionals. However, since this study focuses on nursing students from a specific region, there are limitations in generalizing the results. Additionally, while moral sensitivity emerged as a factor in the AI ethics awareness, it demonstrated a low level of explanatory power, indicating the need to further explore additional factors affecting nursing students’ AI ethics awareness.

## Figures and Tables

**Table 1 healthcare-12-02172-t001:** Participant characteristics and differences in AI ethics awareness (*n* = 140).

Variables	Categories	*n* (%)	Mean ± SD	t or F (*p*)	Scheffé
Gender	Male	9 (6.4)	3.31 ± 0.40	−0.81 (0.439) ^†^	
	Female	131 (93.6)	3.42 ± 0.24		
Grade	1st	12 (8.6)	3.48 ± 0.25	28.44 (0.870) ^§^	
	2nd	66 (47.1)	3.37 ± 0.31		
	3rd	30 (21.4)	3.41 ± 0.26		
	4th	32 (22.9)	3.38 ± 0.35		
Religion	Yes	85 (60.7)	3.33 ± 0.28	−1.98 (0.025)	
	No	55 (39.3)	3.43 ± 0.31		
Experience using AI devices	Yes	105 (75.0)	3.40 ± 0.32	1.00 (0.321)	
	No	35 (25.0)	3.34 ± 0.27		
Experience in ethics education	Yes	130 (92.9)	3.40 ± 0.31	1.48 (0.291)	
	No	10 (7.1)	3.25 ± 0.29		
Experience in AI-related education	Yes	95 (67.9)	3.44 ± 0.32	2.85 (0.003)	
	No	45 (32.1)	3.28 ± 0.29		

^†^ Welch’s *t*-test; ^§^ Welch’s ANOVA.

**Table 2 healthcare-12-02172-t002:** Degrees of digital literacy, moral sensitivity, and AI ethics awareness (*n* = 140).

Variables	Mean ± SD ^†^	Min	Max	Possible Range
Digital Literacy (18 items)	4.09 ± 0.73	3.34	5.00	1–5
Moral Sensitivity (36 items)	4.25 ± 0.30	3.15	4.86	1–5
Respect for patients (6)	4.39 ± 0.83	2.05	5.00	1–5
Professional ethics (6)	4.31 ± 0.32	0.00	5.00	1–5
Responsibility for nursing work (6)	4.35 ± 0.37	0.00	5.00	1–5
Willingness to do good (5)	4.09 ± 0.35	0.00	5.00	1–5
Ethical deliberation (4)	4.20 ± 0.36	2.04	5.00	1–5
Ethical burden (3)	4.21 ± 0.47	1.00	5.00	1–5
Ethical situation awareness (3)	4.13 ± 0.45	1.71	5.00	1–5
Empathy (3)	4.10 ± 0.47	1.67	5.00	1–5
AI Ethics Awareness (24 items)	3.39 ± 0.29	2.70	4.24	1–5
Responsibility (3)	3.81 ± 0.34	2.78	5.00	1–5
Stability (3)	2.99 ± 0.29	1.00	5.00	1–5
Non-discrimination (3)	2.34 ± 0.35	1.57	4.00	1–5
Transparency and explainability (3)	4.25 ± 0.32	2.60	5.00	1–5
Human-centered service (3)	3.52 ± 0.32	2.33	5.00	1–5
Employment (3)	4.03 ± 0.37	3.00	5.00	1–5
Allowance and limitations (3)	3.06 ± 0.37	1.67	5.00	1–5
Rights of robots (3)	3.10 ± 0.37	1.00	5.00	1–5

^†^ The value obtained by dividing the total score by the number of items.

**Table 3 healthcare-12-02172-t003:** Correlation among digital literacy, moral sensitivity, and AI ethics awareness (*n* = 140).

Variables	Digital Literacy	Moral Sensitivity	AI Ethics Awareness
	r (*p*)	r (*p*)	r (*p*)
Digital Literacy	1		
Moral Sensitivity	0.59 (<0.001)	1	
AI Ethics Awareness	0.30 (<0.001)	0.27 (<0.001)	1

**Table 4 healthcare-12-02172-t004:** Factors influencing AI ethics awareness (*n* = 140).

Variable	B	SE	*β*	t	*p*
(Constant)	2.13	0.37		5.76	<0.001
Religion ^†^ (No)	0.07	0.06	0.12	1.32	0.109
Experience in AI-related Education ^†^ (Yes)	0.09	0.07	0.12	1.23	0.220
Digital Literacy	0.13	0.09	0.19	1.45	0.149
Moral Sensitivity	0.24	0.12	0.23	2.05	0.042
F = 46.78, R² = 0.16, Adj R² = 0.14, *p* < 0.001					

SE = Standard error; Adj = Adjusted; ^†^ Dummy variable.

## Data Availability

The data sets used and/or analyzed during the current study are available from the corresponding author upon reasonable request.

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
