# Peer review of "Influences of Digital Literacy and Moral Sensitivity on Artificial Intelligence Ethics Awareness Among Nursing Students"

_healthcare, 2024, doi:10.3390/healthcare12212172_

Round 1
Reviewer 1 Report
Comments and Suggestions for Authors
Dear Editor,
Thank you for the invitation to review this manuscript. The manuscript reports the results of the influences of digital literacy and moral sensitivity on artificial intelligence ethics awareness among nursing students. The manuscript is presented well, and essential information about the study is provided clearly. I just have a couple of comments below to enhance the clarity of the manuscript:
- The title is The influence of digital literacy, moral sensitivity. Instead of separating digital literacy and moral sensitivity with a comma (,), I suggest authors replace the comma with "and," so the title will be read as “Influence of digital literacy and moral sensitivity on artificial intelligence ethics awareness among nursing students.”
- Setting: It is informed that the study was conducted in G city and J city, but the country was unknown. Please add the country where the study was conducted. Informing the country will give the readers the bigger picture about the setting in this study.
- The term for one sub-domain of AI ethics is not the same. On page 7, line 247, it is written that the scores were highest for transparency and explanability, followed by employment, accountability, and so on. However, on the method part under the ethic awareness instrument (page 3 lines 140–144) and on Table 2 (page 5), there is no sub-domain of accountability; instead, it is responsibility. I suggest the authors use the same term for the sub-domain of each variable for its consistency.
- In the discussion, the authors mentioned bioethical awareness and suggested future studies to investigate bioethical awareness. However, there is no explanation about bioethical awareness and how it is different from AI ethics awareness, which is the variable of this study. I suggest the authors explain in brief about bioethical awareness and how it is different from AI ethics awareness.
- Please add the strength and limitation of the study.
Author Response
I sincerely appreciate the reviewer's careful advice. It has been revised as shown in the attached table. The revised parts are marked in red in the text.
"Please see the attachment."

Reviewer 2 Report
Comments and Suggestions for Authors
This manuscript contains interesting research content.
Please check the following.
Lines 106-109 show the required sample size calculated using G*power.
I think maybe that the sample size shown here is the number of people required for multiple regression analysis.
Lines 170-173:
The author uses t-tests, ANOVA, Pearson's correlation coefficient, and multiple regression analysis as statistical procedures. The sample sizes required for these statistical procedures are different. Please check and recalculate each procedure.
Please refer to the following explanation.
https://stats.oarc.ucla.edu/other/gpower/one-way-anova-power-analysis/
The term “t-test” is ambiguous. Looking at Table 1, the number of people in each group is unbalanced. Therefore, please check for equal variances and then select the appropriate type of t-test. The results of the calculation are shown below, and we think that Welch's t-test should be used.
Group 1: Sample size = 9
Mean = 3.31
Standard deviation = 0.40
Group 2: Sample size = 131
Mean = 3.42
Standard deviation = 0.24
Test for equality of variances for two groups
F-value = 2.78
Degrees of freedom = (8, 130)
P-value = 0.014 (two-tailed probability)
Since equal variances cannot be assumed (it is necessary to choose the Welch's method),
the following results are obtained.
t-value = 0.81498
Degrees of freedom = 8.40030
P-value = 0.43758 (exact value corresponding to decimal degrees of freedom)
Please also refer to the following.
https://stats.libretexts.org/Workbench/Learning_Statistics_with_SPSS_-_A_Tutorial_for_Psychology_Students_and_Other_Beginners/10%3A_Comparing_Two_Means/10.04%3A_The_Independent_Samples_t-test_(Welch_Test)
Please write "ANOVA" for Analysis of Variance (ANOVA).
In Table 1, the ANOVA found no significant difference. Table 1 indicate "Scheffé", but there is no need for further testing because no significant difference.
However, in the explanation of the statistical processing method, please write that if a significant difference is found in the ANOVA, a further test of Scheffé will be conducted.
In addition, I think the author needs to check for equal variances for ANOVA based on the number of people in Table 1. I think author probably optimal to use Welch's ANOVA. Please see the following explanation.
https://www.statisticshowto.com/welchs-anova/
In Table 2, the items for each scale are different, but how are they processed and displayed in this table?
Digital literacy (18 items), Moral sensitivity (36 items), AI ethics awareness (24 items)
For Table 3, the correlation coefficient between AI Ethics Awareness and AI Ethics Awareness showed 11. Please fix this to 1, as it is 100%.
May I suggest that you re-examine the results and discussion based on the statistical processing corrections?
I hope this is helpful.
Author Response

(The authors gave the same response as above.)

Round 2
Reviewer 2 Report
Comments and Suggestions for Authors
Thank you so much for editing your manuscript based on the comments.
Author explained in the cover letter:
"Table 2 shows the average values of the variables. It also shows the average values by sub-area. The number of questions in the sub-domains has been additionally displayed."
Digital Literacy (18 items); Moral Sensitivity (36 items); AI Ethics Awareness (24 items)
For Digital Literacy (18 items), I assume that the average and standard deviation are calculated by dividing the total score for each question by the number of questions (18). The mean ± standard deviation is 4.09 ± 0.73, and the range is 1-5.
Please add the statistical processing method for the following three variables under Table 2.
Also explain these statistical treatments in the Statistical Analysis section.
For example: Average scores of each variable shows the value obtained by dividing the total score by the number of items.
Line 379:
Please fix the line break for Strength and limitation.
Again, thank you so much.
Author Response
Thank you again for your careful consideration.
- The average score of each variable is the total score divided by the number of items, as indicated in the footnote of Table 2, and this information was also added to the statistical analysis method.
- 6. Strengths and limitations have been revised by changing the lines.